

# Solar wind magnetic holes can cross the bow shock and enter the magnetosheath

Tomas Karlsson[1], Henriette Trollvik[1], Savvas Raptis[1], Hans Nilsson[2], and Hadi Madanian[3]

[1]Division of Space and Plasma Physics, School of Electrical Engineering and Computer Science, KTH Royal Institute of Technology, Stockholm, Sweden
[2]Swedish Institute of Space Physics, Kiruna, Sweden
[3]Laboratory for Atmospheric and Space Physics: Boulder, Colorado, USA

**Correspondence:** Tomas Karlsson (tomask@kth.se)

**Abstract.** Solar wind magnetic holes are localized depressions of the magnetic field strength, on time scales of seconds to minutes. We use Cluster multipoint measurements to identify 26 magnetic holes which are observed just upstream of the bow shock and, a short time later, downstream in the magnetosheath, thus showing that they can penetrate the bow shock and enter the magnetosheath. For two magnetic holes we show that the relation between upstream and downstream properties of the magnetic holes are well described by the MHD Rankine-Hugoniot jump conditions. We also present a small statistical investigation of the correlation between upstream and downstream observations of some properties of the magnetic holes. The temporal scale size, and magnetic field rotation across the magnetic holes are very similar for the upstream and downstream observations, while the depth of the magnetic holes varies more. The results are consistent with the interpretation that magnetic holes in Earth's and Mercury's magnetosheath are of solar wind origin, as has previously been suggested. Since the solar wind magnetic holes can enter the magnetosheath, they may also interact with the magnetopause, representing a new type of localised solar wind-magnetosphere interaction.

## 1 Introduction

Solar wind magnetic holes are localized depressions in the magnetic field strength, on time scales of seconds or minutes. First observed by Turner et al. (1977) at 1 AU, they have since been observed in large parts of the heliosphere (Burlaga et al., 2007; Fränz et al., 2000; Karlsson et al., 2021a; Madanian et al., 2019; Sperveslage et al., 2000; Tsurutani et al., 2002a; Volwerk et al., 2020; Winterhalter et al., 1994; Yu et al., 2021; Zhang et al., 2008a, 2009). Already Turner et al. (1977) noted that magnetic holes could be classified according to how much the magnetic field vector rotated while the magnetic hole crossed the spacecraft. Magnetic holes with little change in the field direction were called 'linear' holes, while those with a considerable rotation were later called 'rotational' magnetic holes by Winterhalter et al. (1994). The two types of magnetic holes probably have different generation mechanisms, but there is no agreement on what those generation mechanisms are. For the rotational magnetic holes, flux annihilation due to (slow) reconnection at the current sheet associated with the magnetic field rotation has been suggested (Turner et al., 1977; Zhang et al., 2008b). For the linear magnetic holes, several generation mechanisms have been suggested. They may be remnants of magnetic mirror mode structures (e.g. Sperveslage et al., 2000; Winterhalter et al.,



1994) or mirror mode structures created when the plasma is marginally mirror unstable (Karlsson et al., 2021a). Other theories

are that the magnetic holes are the result of non-linear interaction of Alfvén waves with the solar wind plasma (Buti et al., 2001; Tsurutani et al., 2002a, b), emerging coherent structures in solar wind turbulence (Perrone et al., 2016; Roytershteyn et al., 2015), or diamagnetic structures formed in the solar corona (Parkhomov et al., 2019).

While isolated magnetic holes in the solar wind have received considerable attention, similar structures in planetary magnetosheaths have not been investigated as much. Note that we here discuss *isolated* magnetic holes, consistent with the original

definition by Turner et al. (1977), both in the solar wind and in the magnetosheath. This is in contrast to mirror mode wave structures. The latter are quasi-periodic magnetic-field depressions, often observed in the magnetosheath, and are believed to be generated locally in the magnetosheath (e.g. Soucek et al., 2008). Karlsson et al. (2015) studied localized density enhancements in Earth's magnetosheath, and observed that one class of such structures were associated with clear magnetic field decreases. They called such structures 'diamagnetic plasmoids' and suggested that they were actually solar wind magnetic holes that had

crossed the bow shock. Similar structures were also found in the Mercury magnetosheath (Karlsson et al., 2016). The increase in density associated with the magnetic field decrease is consistent with the fact that linear solar wind magnetic holes have been showed to be pressure balance structures (e.g. Stevens and Kasper, 2007), where the magnetic field decrease is balanced by an increase in either density or temperature (Volwerk et al., 2020). Magnetic holes observed in the inner coma of comet 67P were also interpreted to be of solar wind origin, and also showed a density increase within the magnetic holes (Plaschke et al.,

2018b).

The hypothesis that isolated magnetosheath magnetic holes are of solar wind origin has some further support. Recently Karlsson et al. (2021a) made a comprehensive study of magnetic holes in the magnetosheath of Mercury and compared them to solar wind magnetic holes near the planet. They found that the statistical distributions of temporal scale sizes, magnetic field rotation across the holes, and depth of the magnetic holes were very similar for the two populations, and suggested that the

magnetosheath magnetic holes were of solar wind origin also for Mercury (Karlsson et al., 2012; Karlsson et al., 2016). In another study, Madanian et al. (2022) showed evidence for crossing of the bow shock by a large-scale upstream magnetic hole by analyzing data from several spacecraft. Finally, Parkhomov et al. (2019) reported on a structure observed in the solar wind that shows considerable similarities to a magnetosheath diamagnetic plasmoid from the observations of Karlsson et al. (2015), with the solar wind observations made about 90 s earlier then the magnetosheath one.

While the above hypothesis may seem reasonable, still solid observational proof is missing, and an alternative hypothesis is that the magnetosheath magnetic holes are created locally, downstream of the bow shock. Since magnetic mirror mode waves are known to be excited in the magnetosheath at times of large ion temperature anisotropy, this would be possible if magnetic hole generation is related to the mirror mode instability in some way, as described above.

The purpose of this paper is to use Cluster multipoint measurements made simultaneously in the solar wind and the down-

stream magnetosheath to identify individual magnetic holes observed by both the upstream and downstream spacecraft, and thus unequivocally show that at least some magnetosheath magnetic holes have a solar wind origin. We first discuss the data and methodology used, and then show a number of examples followed by some statistical properties of the full sample of magnetic holes, before ending with a discussion, summary, and conclusions.





## 2 Data and method

We use data from the FluxGate Magnetometer (FGM) and Cluster Ion Spectrometer (CIS) instruments onboard the four Cluster spacecraft (Balogh et al., 2001; Réme et al., 1997). We have manually identified time periods where at least one spacecraft is located in the solar wind, while at the same time at least one other spacecraft is located in the magnetosheath. The identification of solar wind or magnetosheath plasma is done by inspection of the ion energy flux spectrograms, and ion velocity moments for the spacecraft where these are available, in combination with inspection of the magnetic field strength. This is typically

enough to make an unambiguous determination of the type of region the spacecraft are located in, and the location of the bow shock.

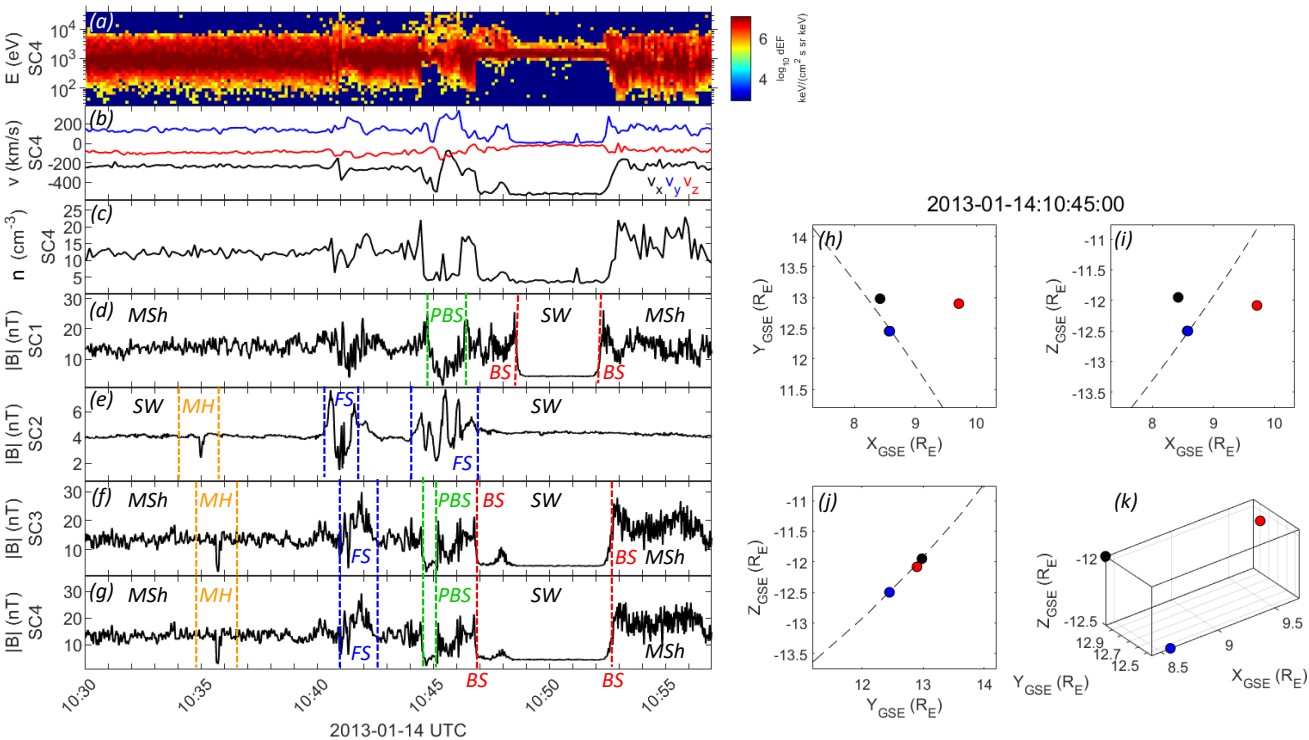

**Figure 1.** Panel (a)-(c): Differential ion energy flux, ion velocity, and ion density for S/C 4, panels (d)-(g): magnetic field strength for S/C 1-4. Identified regions are marked with SW (solar wind), MSh (magnetosheath), MH (magnetic hole), BS (bow shock), FS (foreshock, and magnetosheath downstream of the foreshock), PBS (partial bow shock), see text for further details. Panels (h)-(k): spacecraft positions in various GSE coordinate projections. The spacecraft are identified by the standard Cluster color code; S/C 1 - black, S/C 2 - red, S/C 3 - green, S/4 - blue. S/C 3 and 4 are so close that they cannot be separated on the scale of these plots.

We show an example of such a time interval from 2013-01-14 in Figure 1. Starting with S/C 4, where both magnetic field and ion data are available, we can identify two clear bow shock crossings (marked with 'BS', and dashed red lines) at around 10:46:50 UTC, and 10:52:40 UTC. In between these times we can see a typical solar wind (SW) ion beam, and a low magnetic



field strength of around 4 nT. At later times, and before around 10:41 UTC we can observe the typical heated magnetosheath ion populations, and a compressed magnetic field of around 13-14 nT. During this time the magnetic field has a relatively low level of variability, and a high-energy ion population is not present. This is consistent with the magnetosheath located behind a quasi-perpendicular bow shock, as discussed by Karlsson et al. (2021b). Between 10:41 and 10:42:30 UTC, a region of enhanced magnetic field variability and higher-energy ions can be seen. This is consistent with magnetic field variations and

high-energy particles associated with the foreshock ('FS', here indicating the magnetosheath downstream of the foreshock) of the quasi-parallel bow shock being convected downstream into the magnetosheath, again consistent with the results of Karlsson et al. [2021b]. Just before 10:45 UTC the ion flux data indicates a small, partial excursion into the solar wind, consistent with the decrease in magnetic field strength. We have marked this region 'PBS', for 'partial bow shock crossing'. A similar decrease in magnetic field strength in S/C 3 and 4 has been marked in the same way. In the right part of Figure 1 is shown the spacecraft

positions in GSE coordinates in four different projections at 10:45 UTC. Also indicated is a model bow shock, determined by fitting a paraboloidal model (Merka et al., 2003), using the bow shock position observed by S/C at the crossing taking place at around 10:46:50 UTC. The same method was also used by Karlsson et al. (2021b).

For S/C 2, we see that the magnetic field strength during the whole interval shown is comparable to that observed by S/C 4 when it is located in the solar wind. The magnetic field variability is also very low during almost the whole interval. We

therefore conclude that S/C 2 is located in the solar wind during the whole interval, which is also consistent with the spacecraft position relative to the model bow shock. The time intervals marked with blue dashed lines are associated with some variations in the magnetic field strength, which are likely associated with a foreshock region. This is also consistent with the presence of a high-energy ion population in the downstream magnetosheath during these times. This type of magnetosheath signatures downstream of the foreshock was studied by Karlsson et al. (2021b). We have also estimated the angle $\theta_{Bn}$ between the

normal of the model bow shock (determined at the point where the solar wind intersects the bow shock, assuming its velocity is purely in the GSE $x$ direction) and the magnetic field. $\theta_{Bn}$ is greater than 60° for the whole interval shown, except during the time intervals marked by the blue lines, where it dips down to values below 45°, consistent with the interpretation that these variations are foreshock transients.

S/C 3 is located very close to S/C 4 (it is therefore overplotted in the S/C location plots), and the magnetic field variation

are almost identical to those of S/C 4, meaning that our interpretation of the location of S/C 3 with respect to the bow shock is the same as that for S/C 4. The magnetic field variations of S/C 1 are also very similar, although some differences can be seen due to the slightly larger separation from S/C 4. Still, the general conclusions regarding the S/C 1 position relative to the magnetosheath and solar wind regions remain similar to S/C 4.

This example shows how it is possible to unambiguously identify the position of the spacecraft relative to the bow shock, and

determine whether they are located in the magnetosheath or the solar wind. For our search we concentrate on regions similar to that shown in the beginning of the interval, between 10:30 and 10:40 UTC, where no foreshock structures are observed in the solar wind, in order to easily being able to identify isolated magnetic holes. This also implies that the magnetosheath down-stream of this region, typically associated with the quasi-perpendicular bow shock, is in a less turbulent state, also facilitating identification of magnetic holes there.





During that interval, we can identify an isolated magnetic hole in the solar wind, at around 10:35 UTC, which is also observed in the downstream magnetosheath by S/C 3 and 4, at around 10:35:40 UTC. Below, we will take a closer look at this event, and introduce several other similar observations.

## 2.1 Results

We will begin by presenting two detailed examples of simultaneous observations of magnetic holes in the solar wind and
magnetosheath. After that we will present further examples in less detail, followed by some statistical results from our whole sample.

## 2.2 Example 1, 2013-01-14:10:35 UTC

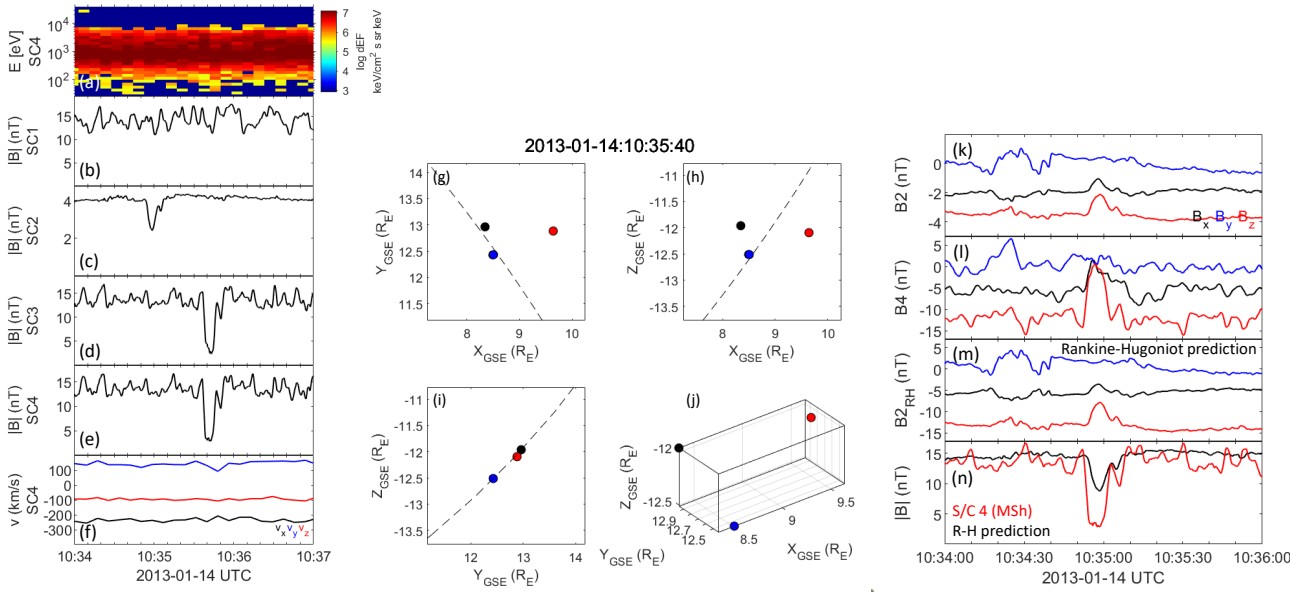

**Figure 2.** Panels (a)-(f): zoomed-in interval of the same event as Figure 1, panels (g)-(j): spacecraft position in the same format as Figure 1, panels (k)-(l): magnetic field components for S/C 2 and 4, in the GSE coordinate system, panel (m): RH prediction for S/C 2, panel (n): RH prediction of magnetic field strength (black), compared with magnetic field strength measured by S/C 4. The data for S/C 2 has been time shifted for easier comparison.

In Figure 2 we show a more detailed view of the magnetic hole shown in Figure 1. Here the magnetic field magnitudes have been smoothed with a 1 s running window in the solar wind (2 s in the magnetosheath), to remove high-frequency variations.
Panels (g)-(j) show spacecraft positions similar to Figure 1, but for a time near the centre of the zoom-in time interval. Panels (k)-(l) show the magnetic field components for the S/C 2 and 4, in the GSE coordinate system. A clear, localized decrease in magnetic field strength, characteristic of magnetic holes can be observed both in the solar wind by S/C 2, and in the





magnetosheath by S/C 3 and 4. The magnetic holes in the solar wind and the magnetosheath have very similar temporal scale sizes. They also have similar, linear polarizations, with most of the magnetic field decrease carried by the $z$ component, less by

the $x$ component, and none by the $y$ component. The magnetic holes also have a clearly identifiable substructure at the trailing edge, clearly seen in S/C 2, 3, and 4. We conclude that the magnetic hole observed by S/C 2-4 is one single entity that has crossed the bow shock and entered the magnetosheath from the solar wind. We note that no similar magnetic hole signature is observed in S/C 1.

We have calculated the relative decrease of the structures by first determining a background magnetic field strenght $B_0$ by

calculating an average of the magnitude of the magnetic field, with a sliding window with a width of 300 s

$$B_0(t) = \langle |\mathbf{B}(t)| \rangle_{300s}, \tag{1}$$

where the angular brackets stand for the averaging operation. We then calculate the relative magnetic field change as

$$\frac{\Delta B}{B_0}(t) = \left\langle \frac{|\mathbf{B}(t)| - B_0}{B_0} \right\rangle_{1s}. \tag{2}$$

We will show time series of $\frac{\Delta B}{B_0}$ below for several events, but for the moment we simply note that the minimum $\frac{\Delta B}{B_0}$ for the

structures observed by S/C 2 is -0.41 and -0.83 and -0.84 for S/C 3 and 4, respectively. We will define an event as a magnetic hole event if a localized magnetic field decrease is below -0.5 in either the solar wind or the magnetosheath region, and there is a similar structure with a decrease of at least -0.4 in the 'complementary' region (in this case the solar wind.) We therefore consider the present example to be a magnetic hole event.

The detailed morphology of magnetic holes are not known, but the fact that the magnetic hole is not observed by S/C 1 indi-

cates that its size in the direction along the separation between S/C 2 and 3 is comparable to that separation length, i.e. around 0.5 $R_E$. We will make a detailed investigation of magnetic holes morphology based on Cluster multi-point measurements in a future study. We can also note that there are no large variations in the ion flow velocity associated with the magnetic hole. This is consistent with the results of Karlsson et al. (2015), who interpreted localized density increases in the magnetosheath correlated with magnetic field decreases as magnetic holes crossing the bow shock. These structures also had no associated

increase in ion flow velocity, and were designated as '(slow) diamagnetic plasmoids'.

In order to study the process of the bow shock crossing in some more detail, we have also compared the downstream magnetic field signatures with the predicted signatures from applying the magnetohydrodynamic (MHD) Rankine-Hugoniot (RH) jump conditions (e.g. Priest, 2012). The RH jump conditions relate downstream and upstream values of the plasma, based on conservation laws in a fluid magnetohydrodynamic description of the plasma. The jump conditions assume a one-

dimensional, time stationary shock, but can be used as a first approximation also in situations that deviate somewhat from these asumptions.

Knowing the upstream conditions, it is possible to solve the RH jump conditions for the downstream fluid parameters. These solutions are most easily expressed in the de Hoffman-Teller (dHT) frame, which is a frame co-moving with the shock in the shock normal direction, having a velocity in the tangential direction chosen so that the upstream magnetic field is parallel to the





upstream plasma flow velocity. (It is easily shown that the magnetic field and flow velocity are then parallel also downstream

of the shock.) In the dHT frame, the RH jump conditions reduce to a two-dimensional problem, and the downstream solutions

can be written as (e.g. Koskinen, 2011; Oliveira, 2017; Priest, 2012)

$$\frac{\rho_d}{\rho_u} = X \tag{3}$$

$$\frac{v_{dn}}{v_{un}} = \frac{1}{X} \tag{4}$$

$$\frac{v_{dt}}{v_{ut}} = \frac{v_u^2 - v_{Au}^2}{v_u^2 - Xv_{Au}^2} \tag{5}$$

$$\frac{B_{dn}}{B_{un}} = 1 \tag{6}$$


$$\frac{B_{dt}}{B_{ut}} = \frac{X(v_u^2 - v_{Au}^2)}{v_u^2 - Xv_{Au}^2} \tag{7}$$

$$\frac{p_d}{p_u} = X + \frac{1}{2}(\gamma - 1)XM_{su}^2 v_u^2 (1 - \frac{v_d^2}{v_u^2}). \tag{8}$$

Here $u$ and $d$ refer to upstream and downstream values, $n$ and $t$ to the normal (to the bow shock) and tangential directions,

$\rho$ is the density, $v$ is the plasma flow velocity, $v_A$ is the Alfvén velocity, $B$ the magnetic field strength, $p$ the pressure, and

$M_s$ the sonic Mach number. The shock compression ratio $X$ is often determined by solving the shock adiabatic equation (e.g.

Priest, 2012). Here we will simply evaluate it from the density or velocity ratios, and use that value to solve for the downstream

magnetic field. We now proceed as follows (a very similar method is used by Keika et al. (2009)):

1. We determine an $lmn$ coordinate system by first fitting a bow shock model to the closest bow shock crossing in the data,

170        as described above. We can then obtain the normal $\hat{\mathbf{n}}$. We let $\hat{\mathbf{l}} = \frac{\hat{\mathbf{z}}_{GSE} \times \hat{\mathbf{n}}}{|\hat{\mathbf{z}}_{GSE} \times \hat{\mathbf{n}}|}$, and let $\hat{\mathbf{m}}$ complete the right-hand system.

2. We transform the flow velocity into the $lmn$ coordinate system, and decompose the velocity in normal and tangential

components:

$$\mathbf{v} = \mathbf{v}_n + \mathbf{v}_t = v_n\hat{\mathbf{n}} + v_t\hat{\mathbf{t}} = v_n\hat{\mathbf{n}} + v_l\hat{\mathbf{l}} + v_m\hat{\mathbf{m}}. \tag{9}$$

3. We transform into the shock frame by subtracting the shock velocity $v_{sh,n}$. This velocity can be determined by observing

175        the upstream and downstream velocities for the closest bow shock crossing ($v_{sh,n} = \frac{[\rho\mathbf{v}]}{[\rho]} \cdot \hat{\mathbf{n}}$, where square brackets denote

the difference between upstream and downstream values.) Thus

$$\mathbf{v}' = \mathbf{v}_n - v_{sh,n}\hat{\mathbf{n}} + \mathbf{v}_t. \tag{10}$$





4. We calculate or determine $X$ (by using the velocities or densities).

5. We determine the deHoffman-Teller velocity for each data point $\mathbf{B}_u$. The dHT velocity is used to transform from the original shock frame to the dHT frame, and is given by (e.g. Kivelson et al., 1995)

$$\mathbf{v}_{HT} = \frac{\hat{\mathbf{n}} \times (\mathbf{v}'_{u,n} \times \mathbf{B}_u)}{\hat{\mathbf{n}} \cdot \mathbf{B}_u}. \tag{11}$$

6. We transform the velocities into the dHT frame for each data point:

$$\mathbf{v}'' = \mathbf{v}' - \mathbf{v_{HT}}. \tag{12}$$

7. We calculate the downstream magnetic field.

8. We transform back into the GSE system.

The result of this procedure is shown in panels (k)-(n) in Figure 2. Here panels (k) and (l) show the magnetic field components in the GSE coordinate system, observed by S/C 2 in the solar wind, and S/C 4 in the magnetosheath. The data for S/C 2 has been shifted by 44 s to facilitate a comparison (also in panels (m)-(n)). Panel (m) shows the solution to the RH jump conditions for the downstream values, based on the upstream values observed by S/C 2, as described above. Calculating the shock velocity according to item 3 yields a shock normal velocity of -1 km/s, which we have used here. Since there is a partial bow shock crossing at 10:45 UTC, the shock velocity is clearly not constant. However, varying it by $\pm$ 25 km/s does not affect the results significantly. We have used the ratio of upstream and downstream number densities observed by S/C 4 to calculate the compression ratio $X$. We can see that there is a reasonably good general agreement between the downstream magnetic field outside of the magnetic hole predicted by the RH jump conditions and the actual downstream values observed by S/C 4. The behaviour of the magnetic field inside the hole is similar in the RH prediction and the S/C 4 observations, described above, with the dominating decrease taking place in the $z$-component, with a smaller decrease in the $x$ component, and no discernible decrease in the $y$ component. We can note that this is an example of a linear magnetic hole. We have calculated the change in magnetic field direction over the magnetic hole by averaging the magnetic field components during 20 s before, and after the magnetic holes, respectively. For the solar wind measurements this gives a change of 3°, and for the magnetosheath measurements 4°. Making the same calculation on the RH-predicted magnetic field yields a rotation of 2°. This is expected, since if the magnetic field direction is similar before and after the magnetic hole observation, the relative change between the normal and tangential components will be the same.

In panel (n) we show the magnitude of the predicted magnetic field compared to the measured downstream values. The general agreement is good, although the RH prediction overestimates the general magnitude somewhat. The prediction also does not reproduce the higher level of downstream fluctuations, which are likely to be generated locally in the magnetosheath. The depth of the magnetic holes is considerably lower for the RH prediction, but the minimum relative change $\frac{\Delta B}{B_0}$ is -0.42, very close to the original solar wind value. This is again expected, since for linear magnetic holes, the direction of the magnetic field vector does not seem to change much over the magnetic hole. Therefore, if we are in the dHT frame, the velocity direction also





does not change much, meaning that the relative change in the normal and tangential components also remains constant. The
mismatch of $\frac{\Delta B}{B_0}$ is likely to be due to either the spacecraft crossing the magnetic hole at different distances from the minimum
magnetic field strength, or changes in the magnetic field configuration during the bow shock crossing not captured by the RH
jump conditions, which, as noted above, are based on assumptions of time stationarity and a one-dimensional geometry. Such
interactions with the bow shock have been suggested by e.g Grib and Leora (2015). This will be discussed further below.

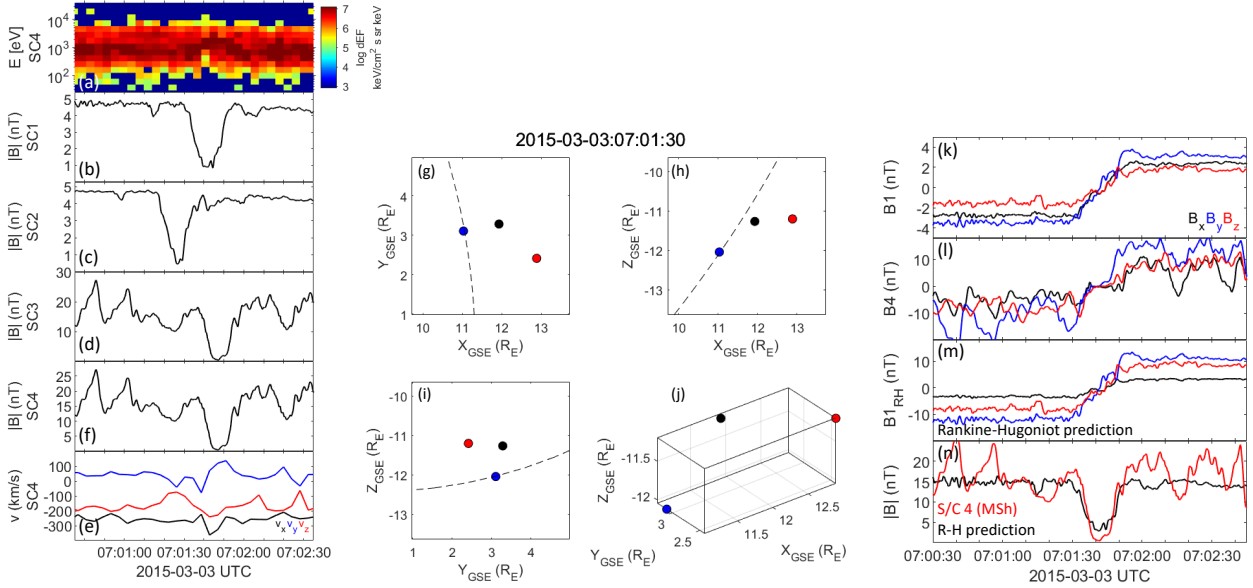

**Figure 3.** Data in same format as Figure 2, but for a rotational magnetic hole from 2015-03-03.

## 2.3 Example 2, 2015-03-03:07:01 UTC

Figure 3 shows a second example of a magnetic hole observed in both the solar wind and the magnetosheath, in the same
format as Figure 2. This time S/C 1 and 2 are situated in the solar wind and the magnetic hole is observed first by S/C 2 and
around 15 s later by S/C 1 consistent with the S/C separation in the $x$ direction. The depths $\frac{\Delta B}{B_0}$ of the magnetic holes are -0.89
and -0.80 for S/C 2 and 1, respectively.

S/C 3 and 4 are located very close to each other, both of them in the magnetosheath, as determined by the wide ion distribution
and the magnetic field magnitude. A magnetic hole is observed by both S/C 3 and 4, around 5 s after the observation by S/C
1. This magnetic hole is of the rotational type, in contrast to the previous example, which can be seen from the magnetic field
components for S/C 1 and 4 shown in panels (k) and (l) of Figure 3. For both spacecraft observations, the magnetic hole is
located at a clear magnetic field rotation/current sheet. The similarities of the magnetic holes relation to the magnetic field
structure between the two spacecraft is further evidence that both spacecraft observe the same magnetic hole. The rotation of
the magnetic field over the magnetic hole is 173° for S/C 1, and 167° for S/C 4.





We have performed a similar Rankine-Hugoniot prediction of the downstream magnetic field as above, based on the S/C 1 data. The results are shown in Figure 3(m)-(n). Again the results are in good general agreement with the actual magnetosheath data from S/C 4. The rotation across the magnetic hole using the RH-predicted magnetic field is 177° verifying that the magnetic field orientation is relatively unchanged by the passage over the bow shock, without showing any signs of a more

complicated interaction with the bow shock, as predicted by e.g. Cable and Lin (1998). This will be discussed again below. The magnitude of the RH-predicted field is somewhat smaller than the S/C 4 observations, and there are large variations in the latter, which are likely due to wave activity generated locally in the magnetosheath.

## 2.4 Further examples

The two examples shown above are strong evidence that the same magnetic holes have been observed both in the solar wind and

the magnetosheath, thus showing that magnetic holes can cross the bow shock, while keeping their basic properties relatively unchanged. In Figure 4 we show a number of further examples, in total 10 different events. For comparison, two of these examples are the events shown above. For all examples the panels show data from two S/C, one in the solar wind and one in the magnetosheath. We show both the magnitude and components of the magnetic field. In addition we show $\frac{\Delta B}{B_0}$ for the two spacecraft in question, with the data from the magnetosheath spacecraft shifted in time for easier comparison. (The time shift

was determined by maximizing the cross correlation between the measurements of the magnetic field magnitude). Comparing subfigures (d) and (g) with the Rankine-Hugoniot predictions from the previous section, we can see that $\frac{\Delta B}{B_0}$ is a reasonable proxy for the RH prediction comparison between the solar wind and magnetosheath measurements. For all examples the detailed agreement between the magnetic field measurements from the solar wind and magnetosheath (scale size, polarization, rotation/linear identity) is strong evidence that magnetic holes cross the bow shock and enter the magnetosheath.

## 245 2.5 Statistics

We have identified in total 26 events of the type shown above. The full list of observation times, together with measured properties of the magnetic holes are given in the dataset (Karlsson et al., 2022). In Figure 5 we show the positions of the full set of magnetic holes observations. We can see that the observations cover a large part of the dayside bow shock, and show a relative good agreement with the statistical bow shock, as evidenced by panel (c).

In Figure 6 we show some statistical results for the full sample of 26 events. In panel (a) we show the rotation across the magnetic holes observed in the magnetosheath versus the rotation of the same magnetic hole in the solar wind. It can be seen that the sample is clearly split in two different populations, one where the solar wind magnetic holes have a rotation less than 40°, and an other where the rotation is greater than 90°. For the purposes of this paper, we will call the latter population rotational magnetic holes, and we indicate this by plotting them in red. This definition is not consistent with some earlier

definitions (Karlsson et al., 2021a, and references therein), but this will not be critical for our conclusions. (The triangular plots symbols will be explained below.) The other population we then call linear magnetic holes, and plot in black. We will use the same color convention in panels (b) and (c). From panel (a) it is clear that even if the correlation between the rotation of the solar wind and magnetosheath observation is not perfect, a rotational magnetic hole generally remains rotational after





passing the bow shock, and the same is true for the linear magnetic holes. The only exception is the outlier with a rotation of
around 150° in the solar wind, but a considerably smaller rotation in the magnetosheath. For this magnetic hole the surrounding
magnetosheath was in a more turbulent state than for the other events, which resulted in a large uncertainty in the determination
of the magnetic field rotation.

Figure 6(b) plots the temporal width $\Delta t$ of each magnetic hole in the solar wind versus the width in the magnetosheath
for the same magnetic hole. $\Delta t$ is defined as the full width at the half minimum. In order to minimize effects of random
fluctuations, we have smoothed the data in the solar wind using a 1 s running window, while in the magnetosheath we have
used a window size of 2 s, to take into account the higher magnetic field variability. For a few magnetic holes the variability in
the magnetosheath was considerably higher than for the other events, and we increased the window size to 4 or 6 s. These data
points are marked with triangles. The window size for each event can be found in the table in the auxiliary material. We can
see that there is a strong correlation between the temporal scale sizes in the magnetosheath and solar wind, indicating that the
magnetic hole temporal scale size is approximately conserved in the crossing of the bow shock. This seems to be true for both
linear and rotational magnetic holes.

Finally, in panel (c) we show the depth of the magnetic field, which we define as the minimum of the ratio $\frac{\Delta B}{B_0}$ for each
magnetic hole. Again we plot the (negative of the) magnetosheath value versus the solar wind one for each magnetic hole.
We have applied the same smoothing as above before determining the depth. Here the spread is large, but it is clear that a
majority of the events fulfill the common definition of a magnetic holes of $\frac{\Delta B}{B_0} < -0.5$ in both regions. Again there is no clear
systematic difference between linear and rotational magnetic holes.

## 3 Discussion

The Cluster multipoint measurements presented here show that both rotational and linear solar wind magnetic holes can cross
Earth's bow shock, while keeping their most important properties relatively unchanged: their general shapes, their magnetic
field rotation, and their temporal scale size. This is consistent with the results in (Karlsson et al., 2021a), where it was shown that
magnetic holes in the solar wind near Mercury and magnetic holes in the Mercury magnetosheath had very similar distributions
of magnetic field rotation and temporal scale sizes. As discussed by Karlsson et al. (2021a), the conservation of the temporal
scale size across the bow shock is consistent with the one-dimensional continuity equation.

In the examples shown here, including the two applications of the Rankine-Hugoniot jump conditions and in the statistical
results discussed above, there is no indication of a more complicated interaction of the magnetic holes and the bow shock, such
as the interaction between directional discontinuities and the bow shock (e.g. Burgess and Schwartz, 1988; Lin, 1997). The
interaction of a tangential discontinuity with the bow shock may, e.g., result in Hot Flow Anomalies (HFAs), (e.g. Schwartz
et al., 2000), which may have quite complicated magnetosheath/downstream signatures, such as a combination of fast and slow
magnetosonic signatures (Eastwood et al., 2008) or magnetosheath jets (Savin et al., 2012), while a rotational discontinuity
may produce a downstream combination of slow and intermediary shocks (Cable and Lin, 1998). For linear magnetic holes,





Grib and Leora (2015) modelled their interaction with the bow shock by considering the magnetic holes as bounded by two tangential discontinuities, and predicted the appearance of a shock wave inside the magnetic hole.

There are a number of possible explanations as to why such complex interactions with the bow shock are not observed here: 1. The magnetic field rotation of the rotational magnetic holes, or the boundaries of the linear magnetic holes, are perhaps not abrupt enough to be considered as discontinuities. 2. The orientation of the current sheets may influence the interaction with the bow shock. It is e.g. known that HFAs are only triggered by tangential discontinuities which have a normal with a large cone angle (Schwartz et al., 2000). 3. HFAs are mainly triggered by tangential discontinuities, while if the rotational magnetic holes are generated by magnetic flux annihilation by reconnection, they are likely to be rotational discontinuities (if they indeed can be considered as discontinuities). 4. There may be a confirmation bias, in that our selection criterion is that the upstream and downstream signatures are similar. Perhaps there are times when rotational solar wind magnetic holes do not penetrate the bow shock in the simple fashion that our observations suggest, but have more complicated downstream/magnetosheath signatures that we have discarded from our selected events. Studying the interaction of both rotational and linear magnetic holes with the bow shock with MHD and hybrid simulations should give further insight into the magnetic hole-bow shock interaction.

The exception to the close upstream and downstream similarities of the magnetic hole properties is the depth of the holes. While the magnetic holes have a $\frac{\Delta B}{B_0}$ of at least -0.4, the correlation between the upstream and downstream values is not as strong as for e.g. the temporal scale size. One explanation could be that the upstream and downstream spacecraft observe different parts of the magnetic hole, and do not probe equally deep into the magnetic holes. This would appear to be inconsistent with the very good temporal scale size upstream-downstream correlation. However, this is not necessarily so. Assume, e.g, that magnetic holes are long cylinders, as suggested by Sundberg et al. (2015) for magnetospheric magnetic holes, and that the magnetic field strength as a function of distance from the minimum of the holes is given by a Gaussian expression. Then it is easily show that the times series of the magnetic field strength for any straight spacecraft path through the hole is given by a Gaussian, only the depth of the hole will be different. Then, as long as the inclination of the orbit to the magnetic field orientation is the same, the scale size as we have defined it (full width at half minimum) will be identically the same, due to geometric similarity. This is of course an idealization, but it shows that the depth of the holes are not necessarily strongly correlated even if the temporal scales are the same. Another possibility is that the determination of the background magnetic field, $B_0$, is affected by the higher magnetic field variability in the magnetosheath compared with the solar wind. A third explanation could be that there actually is some more complicated interaction between the magnetic holes and the bow shock than that implied by one-dimensional MHD (as represented by the RH jump conditions). A further possibility is discussed below.

Linear solar wind magnetic holes typically exhibit a balance between thermal and magnetic pressure, by which we understand that the total pressure is the same inside the magnetic holes as in the outside solar wind plasma (Burlaga and Lemaire, 1978; Madanian et al., 2019; Stevens and Kasper, 2007; Volwerk et al., 2021; Winterhalter et al., 1994). If the magnetic hole plasma fulfills the Rankine-Hugoniot jump conditions, this pressure balance may be disturbed by the bow shock crossing, since the tangential and normal magnetic field components are not transformed in the same way (Equations 6 and 7), and the downstream magnetic field strength therefore depends on $\theta_{Bn}$, while the thermal pressure does not (Equation 8). Immediately



after crossing the bow shock, the plasma inside a linear magnetic hole may therefore not be in pressure balance with its surroundings. This lack of pressure balance may be used as an indication that the magnetic holes are not generated locally in the magnetosheath. The effects of the lack of pressure balance will probably depend on the morphology of the magnetic hole, of which very little is known. If the magnetic holes are elongated along the background magnetic field, as results on diamagnetic plasmoids in the magnetosheath seem to indicate (Karlsson et al., 2012), the pressure dynamics may mainly take place in the magnetic field-aligned direction. Depending on the time scale of these dynamics, this could be an explanation of the difference in the magnetic hole depth between upstream and downstream observations. If the magnetic holes have similar extensions in the directions parallel and perpendicular to the background magnetic field, the pressure dynamics may also take place in the perpendicular direction. This could possibly be related to observations that have been interpreted as expansion or contraction of magnetic holes in the magnetosheath (Yao et al., 2020), although we believe that the uncertainties of these observations are large. We plan to further study the pressure balance of magnetic holes in the magnetosheath, using MMS burst data (Baker et al., 2016), in the near future.

Assuming that the magnetic holes are frozen in to the plasma flow in the solar wind, the fact that the temporal scale sizes are the same in the magnetosheath indicates that the magnetic holes are then also frozen in to the magnetosheath plasma. Some magnetosheath magnetic holes may therefore encounter the magnetopause, and interact with it. If the magnetic holes are associated with a density increase (as for the diamagnetic plasmoids), they will also have a larger dynamic pressure than the surrounding magnetosheath plasma. It can be expected that the magnetic hole–magnetopause interaction can result in similar phenomena as those created by magnetosheath jets, for example triggered localized reconnection, magnetopause surface and compressional waves, impulsive penetration, modified ionspheric flows and even aurora (Plaschke et al., 2018a). This will be the subject of further studies.

## 4 Summary and conclusions

We have used Cluster multipoint measurements to show that both linear and rotational magnetic holes can cross the bow shock and enter the magnetosheath. For the 26 events we have identified, their properties (general shape, temporal scale size, and magnetic field rotation across the hole) are quite unchanged by the passage of the bow shock. The exception is the magnetic field depth, which can vary considerably between the upstream and downstream observations. This may simply be explained by the definition of the scale size, which can be independent of the the magnetic field depth under certain circumstances. It may also possibly be related to the change in relation between the thermal and magnetic pressures, expected from the MHD Rankine-Hugoniot jump conditions. In general, the magnetic holes studied here show no signs of a more complicated interaction with the bow shock than expected by the jump conditions. The results here support the interpretation that isolated magnetic holes found in the magnetosheaths of Earth and Mercury are of solar wind origin, and are not generated locally in the magnetosheath (Karlsson et al., 2012; Karlsson et al., 2015, 2016; Karlsson et al., 2021a). The increased dynamic pressure associated with magnetosheath magnetic holes may interact with the magnetopause in similar ways to magnetosheath jets, which represents a new type of solar wind - magnetosphere interactions that needs to be studied further.



*Data availability.* The data in this study are available via the Cluster Science Archive (Laakso et al., 2010), (https://www.cosmos.esa.int/web/csa).

The full list of observation times, together with measured properties of the magnetic holes are given in the dataset (Karlsson et al., 2022).

*Author contributions.* TK initiated the study, performed the data analysis and wrote the manuscript. All co-authors contributed to the analysis of the results, and with reviewing and editing of the manuscript.

*Competing interests.* No competing interests are present.

*Acknowledgements.* SR and TK are supported by the Swedish National Space Agency (SNSA) grant 90/17. HT and TK are supported by

the Swedish National Space Agency (SNSA) grant 190/19.





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





**Figure 4.** Ten different examples of simultaneous observation of magnetic holes in the solar wind and the magnetosheath. Each event is shown in the same format: magnetic field strength for the solar wind (SW) and magnetosheath (MSh) spacecraft, magnetic field components for the same spacecraft in GSE coordinates, and comparison of $\frac{\Delta B}{B_0}$ for the solar wind (black) and magnetosheath (red) spacecraft. The magnetosheath measurements of $\frac{\Delta B}{B_0}$ have been time shifted.





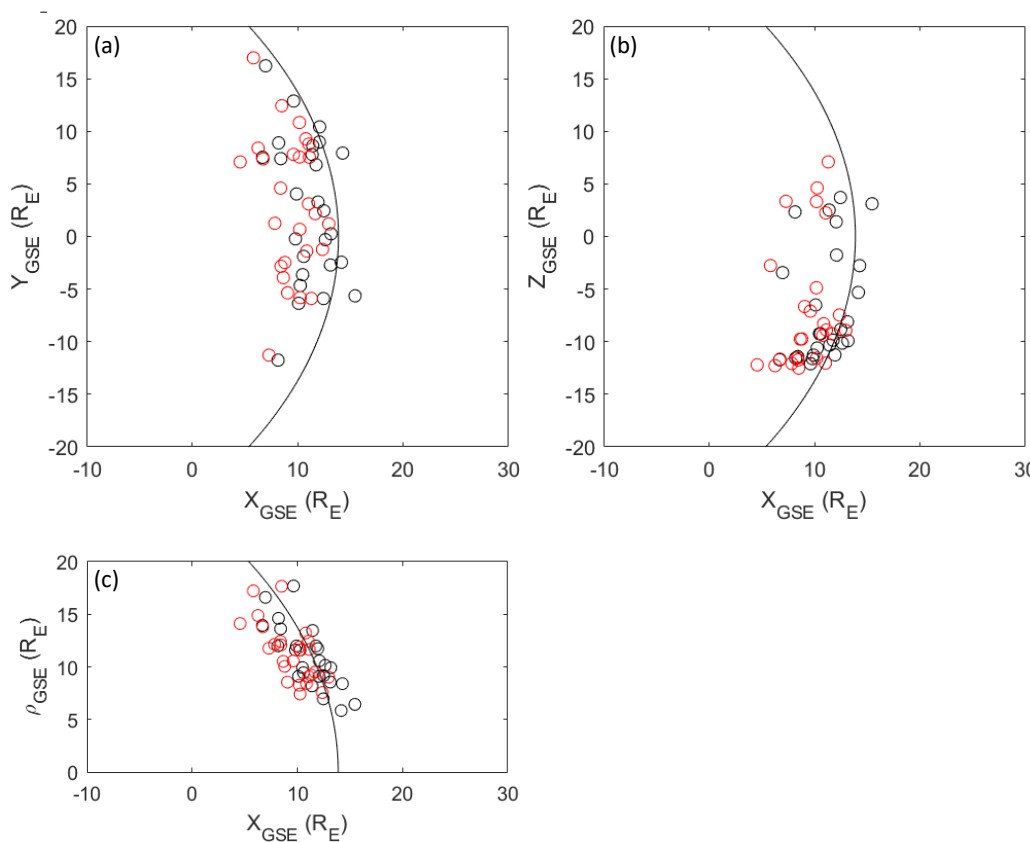

**Figure 5.** Positions of the magnetic holes in GSE $x$-$y$, $x$-$z$, and $x$-$\rho$ projections ($\rho_{GSE} = \sqrt{y_{GSE}^2 + z_{GSE}^2}$). Magnetic holes encountered in the solar wind are shown in black, while magnetosheath observations are marked in red. Also shown is the bow shock position for average solar wind conditions (Kivelson et al., 1995).



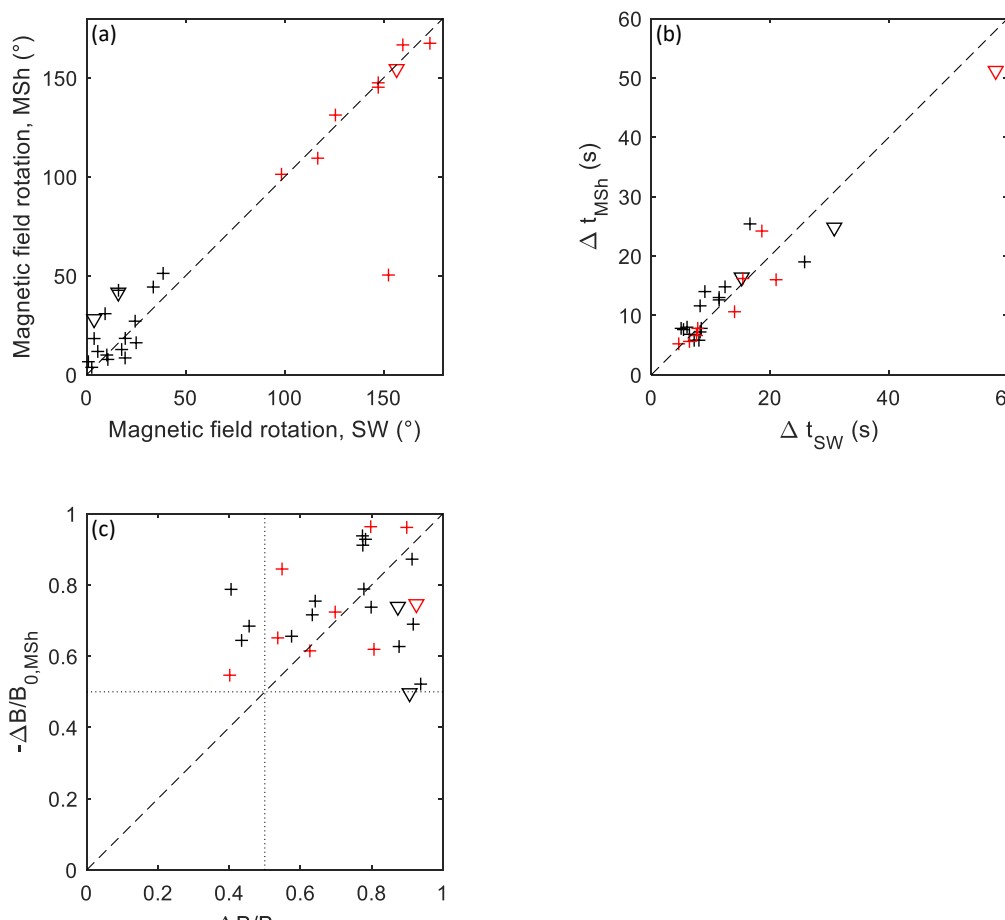

**Figure 6.** Magnetosheath values versus solar wind values for all 26 magnetic holes. Panel (a): magnetic field rotation across the magnetic hole, (b): temporal scale size, (c): magnetic field depth (with the limits of $\Delta B/B_0 = -0.5$ indicated). Black symbols represent linear magnetic holes, and red rotational holes. The triangles indicate heavier smoothing, as described in the main text.