# Peer review of "Solar wind magnetic holes can cross the bow shock and enter the magnetosheath"

_EGUsphere, 2022_

## Author Comment (AC1)

**Reply to Reviewer 1**

We thank the Reviewer for the comment, which we address below. The Reviewer comment is reproduced, and given in italic. Line numbers refer to the original submission.

*These include: (i) hot flow anomalies, which are also characterised by a depressed magnetic field but are filled with hot plasma flowing in a direction significantly deflected from the solar wind velocity vector (e.g. Lucek et al., J. Geophys. Res., doi:10.1029/2003JA010016, 2004); (ii) hot diamagnetic cavities, where the depressed magnetic field within the cavity is flanked by strong enhancements (e.g. Thomsen et al., J. Geophys. Res., doi:10.1029/JA091iA03p02961, 1986); (iii) foreshock cavities, where temperature and pressure inside are only slightly greater than in the ambient solar wind (Sibeck et al., J. Geophys. Res., doi:10.1029/2001JA007539, 2002); and (iv) solar wind density holes, characterised by a strong plasma density depletion within them, flanked by density overshoots and compressed magnetic field (Parks et al., Phys. of Plasmas Lett., 13, 050701, 2006). What is missing from the present paper is putting magnetic holes into context and comparing them with these other transient upstream structures. This is particularly useful for their relationship with density holes, since as shown by Parks et al. (2006) density holes are accompanied by magnetic holes of nearly the same shape.*

The main difference between solar wind magnetic holes and the structures mentioned by the reviewer (to which we may add SLAMS [e.g. *Schwartz and Burgess*, 1991]) is that the latter are strongly associated with the foreshock, and are believed to be generated in or close to this region. In contrast solar wind magnetic holes are known to exist already in the pristine interplanetary solar wind [e.g. *Sperveslage et al.*, 2000]. This is why we have excluded any time intervals where foreshock signatures can be detected, in order to unambiguously identify solar wind magnetic holes (as we originally pointed out). As the reviewer correctly points out, this is particularly important in order to not misidentify density holes (associated with foreshock reflected particles [*Parks et al.*, 2007]) as solar wind magnetic holes. We have extended the discussion starting at line 99 somewhat to clarify the relation of solar wind magnetic holes to the foreshock structures.

---

## Author Comment (AC2)

**Reply to Reviewer 2**

We thank the Reviewer for the comments, which we address below. Reviewer comments are reproduced, and given in italic. Line numbers refer to the original manuscript.

*121-123: This seems to be a suggestion rather than a conclusion. I suggest rephrasing this sentence.*

We assume that the Reviewer means the sentence starting at line 122 (in the original manuscript), since the sentence at line 121 starts with "We conclude". We have simply removed the second sentence.

*Lines 130-134: 'We will define an event as a magnetic hole event if a localized magnetic field decrease is below -0.5 in either the solar wind or the magnetosheath region, and there is a similar structure with a decrease of at least -0.4 in the 'complementary' region (in this case the solar wind.)' These seem to be arbitrary numbers. Is there a physical reason for this choice? Do the results presented in this work vary significantly if the authors consider different values for the magnetic field decrease in both regions?*

A decrease of 50% from the background magnetic field is more or less the standard definition of a magnetic hole, used in many studies before [e.g. *Winterhalter et al.*, 1994; *Sperveslage et al.*, 2000; *Volwerk et al.*, 2020; 2021; *Karlsson et al.*, 2021]. It is logical to adhere to this standard, but we agree that this should have been commented on. We now mention this explicitly. We could of course have demanded this level for both magnetic holes in an upstream-downstream pair, but we felt that this was unnecessarily restrictive, since both the upstream and downstream spacecraft may not sample the holes equally close to their minimum magnetic field. Relaxing this criterion a little bit for one of the magnetic holes increased the already small database a little (five extra events.) We have added a brief discussion on this.

*Figure 3: How do the authors distinguish between magnetic holes (particularly rotational magnetic holes) in the solar wind and current sheet crossings (e.g, heliospheric current sheet crossings)?*

The short answer is that we don't. Rotational magnetic holes may very well be (more or less active) current sheets, as we have already mentioned in the Introduction (line 21). However, earlier results have shown that there is a continuous distribution of rotational angles over the magnetic holes, and only a rather small tail have the really large rotations that would be associated with a crossing of the heliospheric current sheet. Note also that many current sheets would not necessarily be associated with a magnetic field minimum, e.g. a tangential discontinuity with no ongoing reconnection.

*Line 246: The authors stated they identified 26 events of interest. I think it would be worth adding the amount of data that has been analyzed to be able to find them. How often magnetic holes are observed by Cluster?*

The main factor that determines the number of events is the relative rarity of the Cluster spacecraft configurations where one spacecraft is located upstream of the bow shock, while another one is in downstream region. Suitable spacecraft configurations were typically found during February-April during the years 2003, 2006, 2009, 2011, 2013, 2015, 2016, and 2019.

For each orbit, a suitable spacecraft configuration was typically available for less than one hour. This explains the relatively low number of events used in this study. Generally, Cluster is able to observe magnetic holes frequently; *Xiao et al.* [2014] found an occurrence rate of 1.8 per day, also consistent with an occurrence rate of 2 per day based on MMS observations [*Volwerk et al.*, 2021]. We have added a brief discussion on this.

**References**

Karlsson, T., Heyner, D., Volwerk, M., Morooka, M., Plaschke, F., Goetz, C., and Hadid, L.: Magnetic holes in the solar wind and magnetosheath near Mercury, Journal of Geophysical Research: Space Physics, p. e2020JA028961, 2021.

Parks, G. K., Lee, E., Lin, N., Mozer, F., Wilber, M., Lucek, E., ... & Escoubet, P. (2007, August). Density holes in the upstream solar wind. In AIP Conference Proceedings (Vol. 932, No. 1, pp. 9-15). American Institute of Physics.

Schwartz, S. J., & Burgess, D. (1991). Quasi-parallel shocks: A patchwork of three-dimensional structures. Geophysical Research Letters, 18(3), 373-376.

Sperveslage, K., F. M. Neubauer, K. Baumgärtel, N. F. Ness. Magnetic holes in the solar wind between 0.3 AU and 17 AU. Nonlinear Processes in Geophysics, European Geosciences Union (EGU), 2000, 7 (3/4), pp.191-200.

Volwerk, M., Goetz, C., Plaschke, F., Karlsson, T., Heyner, D., and Anderson, B.: On the magnetic characteristics of magnetic holes in the solar wind between Mercury and Venus, in: Annales Geophysicae, vol. 38, pp. 51–60, Copernicus GmbH, 2020.

Volwerk, M., Mautner, D.,Wedlund, C. S., Goetz, C., Plaschke, F., Karlsson, T., Schmid, D., Rojas-Castillo, D., Roberts, O.W., and Varsani, A.: Statistical study of linear magnetic hole structures near Earth, in: Annales Geophysicae, vol. 39, pp. 239–253, Copernicus GmbH, 2021.

Winterhalter, D., Neugebauer, M., Goldstein, B. E., Smith, E. J., Bame, S. J., & Balogh, A. (1994). Ulysses field and plasma observations of magnetic holes in the solar wind and their relation to mirror-mode structures (Paper 94JA01977). Journal of Geophysical Research-Part A-Space Physics-Printed Edition, 99(12), 371-382.

Xiao, T., Shi, Q. Q., Tian, A. M., Sun, W. J., Zhang, H., Shen, X. C., ... & Du, A. M. (2014). Plasma and magnetic-field characteristics of magnetic decreases in the solar wind at 1 AU: Cluster-C1 observations. In Coronal magnetometry (pp. 553-573). Springer, New York, NY.